# An Improved SAMP Algorithm for Sparse Channel Estimation in OFDM System

**DOI:** 10.3390/s23156668

**Published:** 2023-07-25

**Authors:** Hao Hu, Xu Zhao, Shiyong Chen, Tiancong Huang

**Affiliations:** 1School of Microelectronics and Communication Engineering, Chongqing University, Chongqing 400044, China; 2Beijing Smart-Chip Microelectronics Technology Co., Ltd., Beijing 100192, China

**Keywords:** compressed sensing, channel estimation, denoise, step-size adjustment

## Abstract

Channel estimation of an orthogonal frequency division multiplexing (OFDM) system based on compressed sensing can effectively reduce the pilot overhead and improve the utilization rate of spectrum resources. The traditional SAMP algorithm with a fixed step size for sparse channel estimation has the disadvantages of a low estimation efficiency and limited estimation accuracy. An Improved SAMP (ImpSAMP) algorithm is proposed to estimate the channel state information of the OFDM system. In the proposed ImpSAMP algorithm, the received signal is firstly denoised based on the energy-detection method, which can reduce the interferences on channel estimation. Furthermore, the step size is adjusted dynamically according to the *l*_2_ norm of difference between two estimated sparse channel coefficients of adjacent phases to estimate the sparse channel coefficients quickly and accurately. In addition, the double threshold judgment is adopted to enhance the estimation efficiency. The simulation results show that the ImpSAMP algorithm outperforms the traditional SAMP algorithm in estimation efficiency and accuracy.

## 1. Introduction

With the continuous development of mobile communication systems and the rapid rise of emerging technologies represented by the Internet of Things, artificial intelligence, and cloud computing, wireless communication systems are larger and more complex than ever before, which brings many key challenges that cannot be solved by traditional approaches. Among them, the wireless communication signal processing technology is more complex, the channel environment is more variable and complex, and the impact of multipath effect is also worse. Moreover, the demand for spectrum resources is increasing, especially in the wireless band range below 3 GHz, where the frequency demand is very intense. As an important physical resource of wireless communication, wireless spectrum resources have been unable to meet the needs of current and future communication services. According to the investigation of the Federal Communications Commission of the United States, in the allocated frequency spectrum, only a few frequency bands have utilization rates above 10%, and there is a great waste of spectrum resources. The OFDM technology divides the channel into several orthogonal sub-channels to enhance the anti-interference capability. Due to the bandwidth of each sub-channel being smaller than the coherent bandwidth of the channel, the spectral characteristics of each sub-channel are approximately flat. Therefore, it has the ability to resist the multipath effect caused by the complex channel environment, and the mutual orthogonality of each subcarrier greatly improves the utilization rate of spectrum resources, causing OFDM technology to become the key technology of wireless communication [1]. In order to reduce the multipath effect of wireless channels, accurate channel state information (CSI) is required for the OFDM receiver to demodulate user data accurately. Traditional OFDM channel estimation methods do not take into account the sparsity of wireless channels [1,2]. The large number of pilot signals required to detect the time-varying characteristics of the channel results in a waste of spectrum resources. Compressed sensing technology can estimate the channel state information by utilizing the sparsity of wireless channels, which has advantages of saving spectrum resources and reducing pilot overhead [3,4,5,6].

The compressed sensing theory includes three steps: signal sparse representation, signal compression, and signal reconstruction [7]. The signal reconstruction is the key step to compressed sensing theory, which can accurately reconstruct the original high-dimensional data from the low-dimensional compressed data. The performance of signal reconstruction can be evaluated by reconstruction efficiency and accuracy. In recent years, researchers have proposed some signal reconstruction algorithms that can balance the reconstruction efficiency and accuracy; for example, the orthogonal matching pursuit (OMP) algorithm, which was proposed in [8], picked only one best-matching column vector in each reconstruction process, resulting in low reconstruction efficiency. To solve the problem of low efficiency in the OMP algorithm, the GOMP algorithm was proposed in [9], and it selected multiple best-matched column vectors in each reconstruction process to improve the reconstruction efficiency. However, when the number of selected column vectors in each reconstruction process is too large, the number of selected optimal matching column vectors exceeds the actual number of optimal matching column vectors, resulting in a reconstruction error and the degradation of reconstruction accuracy. To solve the problem of column vector over-selection in the GOMP algorithm, the ROMP algorithm was proposed in [10], and it selected firstly multiple best-matching column vectors and then reselected column vectors from the previously selected column vectors based on the regularization constraint criterion, which can effectively solve the over-selection problem in the GOMP algorithm. The efficiency and accuracy of the GOMP algorithm have been effectively improved. All of the abovementioned algorithms require the sparsity of the channel state information as a known parameter for the reconstruction process. However, when the receiver wants to reconstruct the channel state information, it can only analyze the transmitted signal and the received signal. In this process, the receiver does not have any prior information about the state of the transmitted signal after it passes through the wireless channel. Therefore, when the receiver reconstructs the channel state information, the sparsity of the wireless channel cannot be taken as the known parameter. This will have a huge impact on the reconstruction accuracy for OMP, ROMP, GOMP, and other algorithms. In order to avoid this problem and make the reconstruction algorithm not depend on the sparsity of reconstruction signal, the stage-wise orthogonal matching pursuit (STOMP) algorithm, which need not know the sparsity of the signal, was proposed in [11]; the STOMP algorithm used a weak selection strategy that selected all column vectors satisfying a preset threshold in each reconstruction process. However, the threshold setting in the algorithm is based on the norm of the residual between the estimated values and observed values of the sparse signal, and the threshold is a function of the residual, so it results in the selected column vector not always being the best representation of the reconstruction signal. Therefore, the reconstruction accuracy is decreased. To solve the unreasonable threshold setting of STOMP algorithm, the stage-wise weak orthogonal matching pursuit (SWOMP) algorithm was proposed in [12], and it set the threshold based on the inner product between the sensing matrix and the residual of the estimated values and observed values for the sparse signal. Due to the fact that the inner product is a function between the column vector and the residual, it is more favorable to select the column vector. Neither the STOMP algorithm nor the SWOMP algorithm estimate the sparsity of the sparse signal. The SAMP algorithm was proposed to estimate the channel state information [13,14]. In the SAMP algorithm, the true sparsity can be approached gradually by setting the iteration step size. In the column-vector selection process in the SAMP algorithm, the number of column vectors selected is determined by a fixed step size. If the step size is small, it will result in low reconstruction efficiency. If the step size is large, the reconstruction accuracy cannot be guaranteed.

Due to the stop condition of these reconstruction algorithms mentioned above, the residual between the estimated value and the real value is within a certain range, resulting in a low reconstruction efficiency [15]. Moreover, these algorithms do not take into account the existence of noise, thus reducing the estimation accuracy of channel state information. In this paper, an Improved SAMP algorithm by combining the idea of weak selection and the sparsity adaptive is proposed to estimate the sparse channel in the OFDM system. In this algorithm, the received signal is firstly denoised based on energy detection method to improve the anti-noise performance of the algorithm. Different from the SAMP algorithm selecting column vectors according to equal intervals, the proposed algorithm uses a weak selection strategy to select column vectors and improves the efficiency of column-vector selection. Moreover, the step size in the proposed algorithm is determined by the *l*_2_ norm of the difference between the two estimated sparse channel coefficients of adjacent phases, and the double threshold judgment is set to stop the iteration condition, which can improve the estimation efficiency. The simulation results show that the ImpSAMP algorithm performs better than the traditional SAMP algorithm.

## 2. System Model

### 2.1. OFDM Communication System

In the OFDM communication system of this paper, each OFDM symbol is the sum of several modulated subcarriers, and the modulation mode of each subcarrier is quadrature amplitude modulation (QAM). Let each OFDM symbol contain *K* subcarriers, among which there are *M* pilot subcarriers. In order to avoid the inter-symbol interference (ISI) and inter-carrier interference (ICI) generated by the multipath effect, a cycle prefix is inserted into the OFDM symbol. $T_{ofdm}$ and $T_{cp}$ represent the OFDM symbol length and cycle prefix length, respectively. $T=T_{ofdm}+T_{cp}$ is the complete OFDM symbol length. Let $f_{c}$ denote the carrier frequency, and then the frequency of the *k*th subcarrier can be expressed as $f_{k}=f_{c}+k/T,k=0,1,\cdots ,K-1$. The symbol of information transmitted on the *k*th subcarrier is s(k). The *k*th subcarrier after OFDM modulation can be expressed as
(1)$$x(t)=s(k)e^{j2\pi f_{k}t},\;0\leq t\leq T.$$

If the subcarrier passes through the frequency-selective fading channel, the channel impulse response, h(τ), can be expressed as
(2)$$h(\tau )=\sum_{p=1}^{N_{p}}A_{p}\delta (\tau -\tau_{p}),$$
where $A_{p}$ denotes the energy coefficient, *τ_p_* denotes the path delay of the *p*th path, and *N_p_* denotes the total number of paths.

At the receiving end, the transmitted *k*th subcarrier, x(t), after passing through the channel, can be denoted as follows:(3)$$y(t)=x(t)*h(t)+v(t)=\sum_{p=1}^{N_{p}}A_{p}x(t-\tau_{p})+v(t),\;0\leq t\leq T,$$
where v(t) is the Additive White Gaussian Noise (AWGN).

In the frequency domain, the output of the *k*th subcarrier after passing through the channel can be expressed as
(4)Y(k)=X(k)H(k)+V(k),
where X(k) denotes the *k*th transmitted subcarrier, V(k) denotes the Gaussian white noise added to the *k*th subcarrier, and H(k) denotes the channel frequency response. Through the Fourier transform, H(k) can be expressed as
(5)$$H(k)=ℱ[h(\tau )]=\sum_{p=1}^{N_{p}}A_{p}e^{-j2\pi \frac{k}{T}\tau_{p}},\;k=0,1,\ldots ,K-1.$$

### 2.2. OFDM Channel Estimation Model Based on CS Theory

The compressed sensing (CS) theory is widely used in image processing [16], radar detection [17], channel estimation [18,19,20,21,22], and other signal processing fields. The CS theory shows that the signal is compressible if it satisfies the sparse condition, and the high-dimensional sparse signal can be projected into the low-dimensional signal by the transform basis, which is not related to the observation matrix. The original signal can be accurately constructed from a very small number of projected signals, where sparse condition means that the value of most points of the signal in the transformation domain is zero and only a few points are non-zero, so the signal is sparse in this transformation domain, where the number of non-zero points is called the sparsity of the signal.

If *M* pilot subcarriers are used for channel estimation, the position of the pilot subcarriers in the subcarriers is denoted as $P_{M}=\left(p_{1},p_{2},\cdots ,p_{M}\right)$; $Y_{M}={\left[y(p_{1}),y(p_{2}),\ldots ,y(p_{M})\right]}^{T}$ denotes the received pilot subcarriers at the pilot position $P_{M}$; and $X_{M}=diag(x(p_{1}),x(p_{2}),\ldots ,x(p_{M}))$ denotes the transmitted pilot subcarriers at the pilot position, $P_{M}$. It can be seen from Equation (4) that the output of *M* pilot subcarriers can be expressed as
(6)$$Y_{M}=X_{M}H_{M}+V_{M},$$
where $V_{M}={\left[v(p_{1}),v(p_{2}),\ldots ,v(p_{M})\right]}^{T}$ denotes the AWGN at the pilot position, $P_{M}$; and $H_{M}$ denotes the channel frequency response at the pilot position, $P_{M}$.

The channel estimation model of the OFDM communication system based on the compressed sensing theory is shown in Figure 1. It can be seen that the OFDM channel estimation can be transformed into solving the sparse channel coefficients, θ. It is assumed that the channel matrix, $H_{M}$, satisfies the sparse condition, and the channel matrix, $H_{M}$, can be expressed as $H_{M}=F_{M}\theta$. The entire observation process is that the sparse channel coefficients, θ, are observed by the sensing matrix, A. Therefore, the observation vector, $Y_{M}$, can be represented as
(7)$${\left(Y_{M}\right)}_{M\times 1}=A_{M\times N}\theta_{N\times 1},M<N.$$

The compressed sensing model shown in Equation (7) is a problem of solving the underdetermined equation about the sparse channel coefficients, θ. Theoretically, there are infinite solutions, and the sparse channel coefficients (θ) cannot be precisely estimated. When the sparsity of θ is less than *M*, the sparsest solution that satisfies $Y_{M}=A\theta$ is the desired one [23]. Therefore, the problem can be described to minimize the *l*_0_ norm of θ as follows:(8)$$\hat{\theta }=\underset{\theta }{\arg\min}{\left\|\theta \right\|}_{0}\;s.t.\;Y_{M}=A\theta .$$

In fact, it is very difficult to find the sparsest solution by satisfying $Y_{M}=A\theta$. The sparse channel coefficients, θ, can be estimated by the observed vector, $Y_{M}$, and its corresponding estimated vector, $\hat{Y_{M}}$. The residual, r, is defined as the difference between the observed vector, $Y_{M}$, and the estimated vector, $\hat{Y_{M}}$. The square sum of r between the observed vector value, $Y_{M}$, and the estimated vector value, $\hat{Y_{M}}=A\hat{\theta }$, is minimized and can be described as follows
(9)$$\hat{\theta }=\underset{\theta }{\arg\min}{\left\|Y_{M}-\hat{Y_{M}}\right\|}_{2}^{2}.$$

The square sum of r can be expressed as follows:(10)$$\begin{array}{cc} E\left(r\right) & =r^{T}r\\ & ={\left(Y_{M}-\hat{Y_{M}}\right)}^{T}\left(Y_{M}-\hat{Y_{M}}\right)\\ & =Y_{M}{}^{T}Y_{M}-{\hat{\theta }}^{T}A^{T}Y_{M}-Y_{M}{}^{T}A\hat{\theta }+{\hat{\theta }}^{T}A^{T}A\hat{\theta }. \end{array}$$

The partial derivative of E(r) to $\hat{\theta }$ can be expressed as follows:(11)$$\begin{array}{cc} \frac{\partial E}{\partial \hat{\theta }} & =\frac{\partial }{\partial \hat{\theta }}\left(Y_{M}{}^{T}Y_{M}-{\hat{\theta }}^{T}A^{T}Y_{M}-Y_{M}{}^{T}A\hat{\theta }+{\hat{\theta }}^{T}A^{T}A\hat{\theta }\right)\\ & =-Y_{M}{}^{T}A+{\hat{\theta }}^{T}A^{T}A. \end{array}$$

Let $\frac{\partial E}{\partial \hat{\theta }}=0$, and the obtained $\hat{\theta }$ that minimizes the square sum of r can be described as
(12)$$\hat{\theta }={\left(A^{T}A\right)}^{-1}A^{T}Y_{M},$$
where $\hat{\theta }={\left(\hat{\theta_{1}},\hat{\theta_{2}},\cdots ,\hat{\theta_{j}},\cdots ,\hat{\theta_{N}}\right)}^{T}$ is an *N* × 1 column vector. If the $\hat{\theta }$ in Equation (12) is estimated, the *L* terms with the largest absolute value among the terms $\hat{\theta_{j}}$ of θ are selected to form the vector ${\hat{\theta }}_{L\times 1}$, and the column vectors ($a_{j}$) corresponding to the *L* terms with the largest absolute value among the terms $\hat{\theta_{j}}$ of $\hat{\theta }$ are added to the set $A_{M\times L}$. The estimated vector value, $\hat{Y_{M}}$, can be represented as $\hat{Y_{M}}=A_{M\times L}{\hat{\theta }}_{L\times 1}$. The corresponding residual r can be described as follows:(13)$$r=Y_{M}-A_{M\times L}{\hat{\theta }}_{L\times 1}.$$

By substituting (12) into (13), the residual r can be written as follows:(14)$$r=Y_{M}-A_{M\times L}{\left(A_{M\times L}{}^{T}A_{M\times L}\right)}^{-1}A_{M\times L}{}^{T}Y_{M}.$$

## 3. The Proposed Improved SAMP Algorithm

### 3.1. The Improved SAMP Algorithm

The traditional SAMP algorithm gradually increases the number of column vectors with a fixed step size to approximate the true sparsity when the actual sparsity is unknown. If too many preselected column vectors are selected, it may result in over-selection or even wrong selection, which leads to the low accuracy of the channel estimation. Secondly, if the step size chosen is too small, the channel estimation will be less efficient, although the accuracy of the channel estimation can be guaranteed. On the other hand, if the step size chosen is too large, the channel estimation accuracy cannot be guaranteed, though it can be more efficient.

To solve the above problems existing in the traditional SAMP algorithm, the ImpSAMP algorithm is proposed in this paper. The main idea of the proposed ImpSAMP algorithm is firstly to process the noise contained in the received signal based on the energy detection method. Moreover, the weak selection strategy is adopted to select column vectors to improve the efficiency of the column-vector selection. The step size in the proposed algorithm is adjusted dynamically according to the *l*_2_ norm of difference between two estimated sparse channel coefficients of adjacent phases. The double threshold judgment is introduced to improve the channel estimation efficiency.

#### 3.1.1. Denoising Processing

Noise, as a part of the received signal, is also involved in sparse channel estimation, which will reduce estimation accuracy. Therefore, it is necessary to denoise the signal at the receiving end.

The received signal $Y_{M}={\left(y_{1},y_{2},\cdots ,y_{M}\right)}^{T}$ is a column vector. $Y_{M}$ can be converted into an *M* × *M* Hankel matrix, B, which is shown in Equation (15):(15)$$B_{M\times M}=\left[\begin{array}{cccc} y_{1} & y_{2} & \cdots & y_{M}\\ y_{2} & y_{3} & \cdots & y_{M+1}\\ \vdots & \vdots & \vdots & \vdots \\ y_{M} & y_{M+1} & \cdots & y_{2M-1} \end{array}\right].$$

If the Hankel matrix, B, can be decomposed by the singular method, the Hankel matrix, B, can be represented as
(16)$$B=USV^{H},$$
where U and V are *M* × *M* unitary matrices. U and V can be denoted as $U=\left(u_{1},u_{2},\cdots ,u_{j},\cdots ,u_{M}\right)$ and $V=\left(v_{1},v_{2},\cdots ,v_{j},\cdots ,v_{M}\right)$, respectively, where $u_{j}$ and $v_{j}$ are *M* × 1 column vector. The singular matrix $S=diag(s_{11},s_{22},\cdots ,s_{MM})$ is a diagonal matrix whose main diagonal elements are positive and can be arranged according to the following rules:(17)$$s_{11}\geq s_{22}\geq \cdots s_{MM}>0.$$

These singular values contain all useful information of the received signal, where the larger singular value represents the useful signal, and the smaller singular value represents the noise. In order to eliminate the influence of noise on the received signal, it is necessary to retain the larger singular values and delete the smaller singular values appropriately. Due to the fact that the power of the useful signal is significantly higher than the power of the noise signal, the idea of energy detection is used to distinguish the useful signal from the noise signal according to the singular values [24].

The square sum of all singular values can be described as follows:(18)$$P=\sum_{n=1}^{M}{\left|s_{nn}\right|}^{2}.$$

The square sum of the previous *i* singular values can be expressed as follows:(19)$$P_{i}=\sum_{n=1}^{i}{\left|s_{nn}\right|}^{2},1\leq i\leq M.$$

The $P_{i}$ is compared with the preset threshold, β⋅P. If the $P_{i}$ is no less than the β⋅P, this indicates that the previous *i* singular values denote the singular values of useful signal and that these singular values should be kept, and the rest of the singular values that represent the noise signal in the received signal can be set as 0. The singular matrix, S, can be updated as follows:(20)$$S{}^{\prime }=diag(s_{11},s_{22},\cdots ,s_{ii},0,\cdots ,0).$$

The corresponding unitary matrices, U and V, are updated and represented as follows:(21)$$U{}^{\prime }=\left(u_{1},u_{2},\cdots ,u_{i},0,\cdots ,0\right),V{}^{\prime }=\left(v_{1},v_{2},\cdots ,v_{i},\cdots ,0\right).$$

From Equation (16), the corresponding Hankel matrix, $B{}^{\prime }$, can be rewritten as follows:(22)$$B{}^{\prime }=U{}^{\prime }S{}^{\prime }V{{}^{\prime }}^{H}=\left[\begin{array}{cccc} y_{1}{}^{\prime } & y_{2}{}^{\prime } & \cdots & y_{M}{}^{\prime }\\ y_{2}{}^{\prime } & y_{3}{}^{\prime } & \cdots & y_{M+1}{}^{\prime }\\ \vdots & \vdots & \vdots & \vdots \\ y_{M}{}^{\prime } & y_{M+1}{}^{\prime } & \cdots & y_{2M-1}{}^{\prime } \end{array}\right].$$

The denoised signal can be represented as $Y_{M}{}^{\prime }={\left(y_{1}{}^{\prime },y_{2}{}^{\prime },\cdots ,y_{M}{}^{\prime }\right)}^{T}.$

#### 3.1.2. Weak Selection Strategy

It is assumed that the sensing matrix, A, is represented by *N* column vectors as $A=\left(a_{1},a_{2},\cdots ,a_{j},\cdots ,a_{N}\right)$. According to Equation (9), the sparse channel coefficients, θ, can be estimated. The traditional SAMP algorithm uses a fixed step size to estimate the θ. In the column-vector ($a_{j}$) selection stage, the number of column vectors ($a_{j}$) selected in the SAMP algorithm is determined by the step size. If the step size is small, it will result in low estimation efficiency. If the step size is large, the estimation accuracy cannot be guaranteed. The proposed algorithm uses a weak selection strategy to select the column vector, $a_{j}$, by setting the fuzzy threshold, λ, to determine the number of column vectors ($a_{j}$) selected. If the absolute value of the product $\left|a_{j}{}^{T}r\right|$ is greater than λ, the column number, *j*, of the column vector, $a_{j}$, is added to the set $S_{k}$. Thus, we obtain the following:(23)$$S_{k}=\left\{j|u=abs\left(a_{j}{}^{T}r\right)>\lambda ,\;j=1,2,\ldots ,N\right\},$$
where the fuzzy threshold is set as $\lambda =\alpha^{*}\max\left(abs\left(a_{j}{}^{T}r\right)\right)$, and the fuzzy threshold coefficient is α∈(0,1). In the column-vector ($a_{j}$) selection stage, using the weak selection strategy causes the number of column vectors ($a_{j}$) selected to not be limited by the step size, and the number of column vectors ($a_{j}$) selected can be adjusted dynamically according to the absolute value of the product $\left|a_{j}{}^{T}r\right|$, which changes the selection method and the number of column vectors ($a_{j}$) selected, effectively avoiding the problems of too many selections and selection errors, enhancing the selection efficiency, and improving the estimation accuracy of the algorithm. Thus, the algorithm proposed in this paper adopts the weak selection strategy.

#### 3.1.3. The Double Threshold Judgment

In this paper, the step size is changed according to the *l*_2_ norm of the difference between two estimated sparse channel coefficients, $\hat{\theta }$, of adjacent phases. The threshold value, $\varepsilon_{1}$, is set as the stop iteration condition 1, the threshold value $\varepsilon_{2}$ is set as the stop iteration condition 2, and $\varepsilon_{2}<\varepsilon_{1}$. According to (10), (11), (12), and (13), the channel coefficients ($\hat{\theta }$) are continuously estimated. The *l*_2_ norm of difference between the *t*th and (*t*−1)th estimated value for $\hat{\theta }$ is denoted as ${\left\|{\hat{\theta }}_{t}-{\hat{\theta }}_{t-1}\right\|}_{2}$. If ${\left\|{\hat{\theta }}_{t}-{\hat{\theta }}_{t-1}\right\|}_{2}\leq \varepsilon_{1}$ and ${\left\|{\hat{\theta }}_{t}-{\hat{\theta }}_{t-1}\right\|}_{2}\leq \varepsilon_{2}$, this indicates that the estimated channel coefficients ($\hat{\theta }$) are approximately equal to θ, and the process of the estimated channel coefficients is stopped. If ${\left\|{\hat{\theta }}_{t}-{\hat{\theta }}_{t-1}\right\|}_{2}\leq \varepsilon_{1}$ and ${\left\|{\hat{\theta }}_{t}-{\hat{\theta }}_{t-1}\right\|}_{2}>\varepsilon_{2}$ are satisfied, the step size is updated to half of the current step size to improve the estimation accuracy by using a smaller step size. Furthermore, if ${\left\|{\hat{\theta }}_{t}-{\hat{\theta }}_{t-1}\right\|}_{2}>\varepsilon_{1}$, the step size remains unchanged in order to improve the estimation efficiency.

Due to the fact that the threshold value has a close relationship with the observation vector length, *M*; the sparse channel coefficients’ length, *N*; the step size, *S*; and the *l*_2_ norm of the estimated channel coefficients, ${\left\|{\hat{\theta }}_{t}\right\|}_{2}$, the algorithm performance is better when $\varepsilon_{1}$ is taken as $\mathrm{lg}(S){\left(N/M\right)}^{4}e^{-5}{\left\|{\hat{\theta }}_{t}\right\|}_{2}$ and $\varepsilon_{2}$ is taken as $0.2\varepsilon_{1}$ according to the analysis of a large number of experimental results [25].

The detailed process of this algorithm is shown in Figure 2, where *t* denotes the number of iterations, ∅ denotes the empty set, *L* denotes the estimated sparsity, and ***Λ*** denotes the set of column numbers (*j*) of the column vector ($a_{j}$) selected in each iteration.

## 4. Simulation and Experimental Results

For the OFDM communication system, this section verifies the performance of the proposed Improved SAMP algorithm. In this simulation system, the total number of OFDM subcarriers is 256, the number of pilots is 100, and the pilot is inserted randomly. The channel model adopts the Rayleigh fading channel, and the channel length and sparsity are 256 and 15, respectively. The main parameters of the simulation experiment are set as shown in Table 1.

In order to further compare the performance of the proposed Improved SAMP algorithm with other reconstruction algorithms, the mean square error (*MSE*) and bit error rate (*BER*) are used as evaluation criteria in this paper.

The *MSE* is used to describe the error between $\hat{\theta }$ and θ. The expression of *MSE* is
(24)$$MSE=\frac{1}{N}{\sum_{i=1}^{N}\left({\hat{\theta }}_{i}-\theta_{i}\right)}^{2},$$
where *N* denotes the number of simulations, which is set as 1000 in this paper.

The bit error rate (*BER*) describes the probability that a bit is transmitted incorrectly in the communication transmission system. The expression of *BER* is
(25)$$BER=\frac{\sigma }{Z},$$
where σ denotes the number of received bits in error, and *Z* denotes the total number of transmitted bits.

Figure 3 analyzes the effect of different denoising algorithms on the channel estimation mean square error performance in the range of 0 dB to 9 dB for the SNR. The ImpSAMP-With-Noise algorithm processes the received signal directly, without the denoising process. The ImpSAMP-mean algorithm uses the mean-value method to distinguish the useful signal from the noise signal, while the proposed ImpSAMP algorithm distinguishes the useful signal from the noise signal based on the energy detection method. From Figure 3, it can be seen that the ImpSAMP algorithm has a better denoising effect than the ImpSAMP-mean algorithm when the SNR is varied from 0 dB to 4 dB. When the SNR is varied from 5 dB to 9 dB, the difference of the denoising process for two algorithms is not obvious. It can be interpreted that the noise has little effect on the MSE because the SNR is high enough.

Figure 4 analyzes the effect of energy threshold coefficient, β, on the MSE. It can be seen that the MSE gradually decreases as the SNR increases. A large number of experiments have proved that when the energy threshold coefficient, β, is set as 0.1–0.6, a large number of useful signals are removed as noise, and the signal energy is low, resulting in the fuzzy threshold value set being too small. Then, the number of column vectors, $a_{j}$, selected according to the fuzzy threshold value is too high in each reconstruction process, thus causing the set $S_{k}$ to overflow easily, and the algorithm cannot run properly, so the energy threshold coefficient, β, is taken to be in the range of 0.7 to 1.0 in this experiment. It can be seen from Figure 4 that the MSE is better at the same SNR if the energy threshold coefficient, β, is smaller in the range of 0.7 to 1.0. Thus, the energy threshold coefficient, β, is set as 0.7 in the simulation experiment.

Figure 5 shows that the MSE is affected by the fuzzy threshold coefficient, α. It can be seen that the MSE gradually decreases with the increase of the SNR. If the value of the fuzzy threshold coefficient, α, is set as 1.0, the judgment condition, Equation (23), becomes $abs\left(a_{j}{}^{T}r\right)>\max\left(abs\left(a_{j}{}^{T}r\right)\right)$, and no valid column vector can be selected at this time. If the fuzzy threshold coefficient, α, is selected from 0.1 to 0.3, it has a better estimation effect. The proposed fuzzy threshold coefficient, α, is 0.1 or 0.2. Thus, the fuzzy threshold coefficient, α, was set as 0.2 in the simulation experiment of this study.

Figure 6 shows the MSE of channel estimation of the proposed ImpSAMP, ImpSAMP-With-Noise, SAMP, STOMP, and SWOMP algorithms for the Eb/N0 in the range from 0 dB to 30 dB, respectively. The ImpSAMP-With-Noise algorithm processes the received signal directly without denoising process. By comparing the proposed ImpSAMP algorithm with the ImpSAMP-With-Noise algorithm, it can be seen that the denoising process can enhance the channel estimation performance if the Eb/N0 is in the range of 0 dB to 5 dB. If the Eb/N0 is in the range of 5 dB to 30 dB, the channel estimation MSE is not improved obviously by the denoising process. This is because the useful signal power is significantly larger than the noise power, and the noise has little effect on the channel estimation MSE. Compared with other algorithms, the proposed ImpSAMP algorithm improves the estimation accuracy and efficiency of the denoising process and dynamically adjusts the step size according to the *l*_2_ norm of the difference between two estimated sparse channel coefficients of adjacent phases.

Figure 7 shows the BER of the proposed ImpSAMP, the ImpSAMP-With-Noise, SAMP, STOMP, and SWOMP algorithms for the Eb/N0 in the range of 0 dB to 30 dB. The simulation results show that the corresponding BER of the five algorithms mentioned above show the same overall trend: they all show a decreasing trend with the increase of the Eb/N0. The proposed ImpSAMP algorithm can obtain a better BER at a lower Eb/N0 compared with other algorithms.

Figure 8 shows the running time of the proposed ImpSAMP, ImpSAMP-With-Noise, SAMP, STOMP, and SWOMP algorithms for the SNR in the range from 0 dB to 9 dB. Compared with ImpSAMP-With-Noise algorithm, the proposed ImpSAMP algorithm has a longer running time because it contains a denoising process. Compared with the SAMP algorithm, the proposed ImpSAMP algorithm has a shorter running time because the step size of the proposed ImpSAMP algorithm is adjusted dynamically according to the *l*_2_ norm of the difference between two estimated sparse channel coefficients of adjacent phases. The running time of the proposed ImpSAMP algorithm is longer than that of the STOMP and SWOMP algorithm. This is because the proposed ImpSAMP algorithm improves the estimation accuracy by dynamically adjusting the step size and the number of column vectors selected, leading to a longer running time.

## 5. Conclusions

In this paper, by combining the idea of the weak selection of the STOMP algorithm and the sparsity adaptivity of the SAMP algorithm, an ImpSAMP algorithm is proposed to estimate the sparse channel in the OFDM communication system. Compared with the traditional SAMP algorithm, the denoising process based on the idea of energy detection in the proposed ImpSAMP algorithm is adopted to improve channel estimation accuracy. At the same time, the weak selection is adopted to select the column vector to improve the efficiency of the column-vector selection. In addition, the proposed ImpSAMP algorithm adjusts the step size according to the *l*_2_ norm of the difference between two estimated sparse channel coefficients of adjacent phases, and double threshold judgment is set to balance the estimation efficiency and accuracy. The simulation results show that the proposed ImpSAMP algorithm has a lower mean square error of channel estimation and lower BER compared with other reconstruction algorithms.

## Figures and Tables

**Figure 1 sensors-23-06668-f001:**
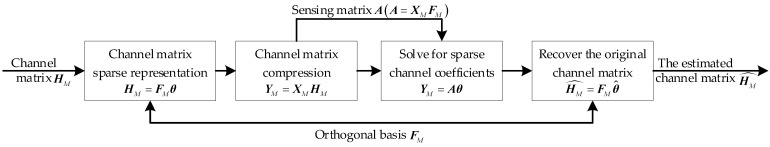
Channel estimation based on CS theory.

**Figure 2 sensors-23-06668-f002:**
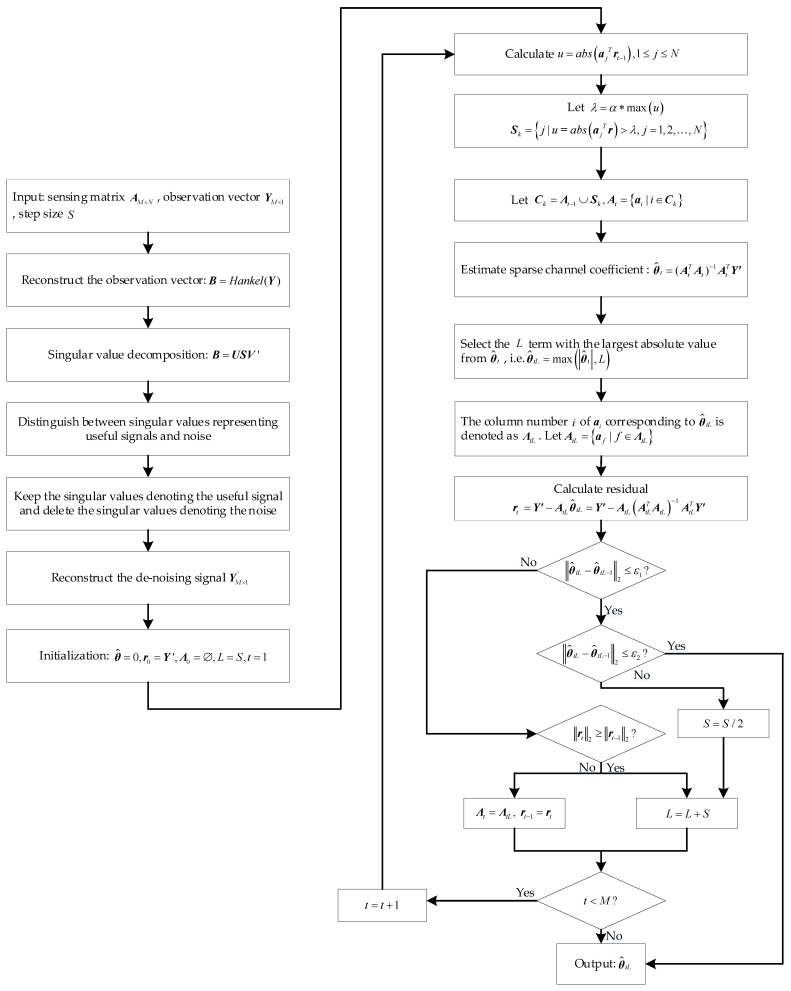
Flowchart of the proposed ImpSAMP algorithm.

**Figure 3 sensors-23-06668-f003:**
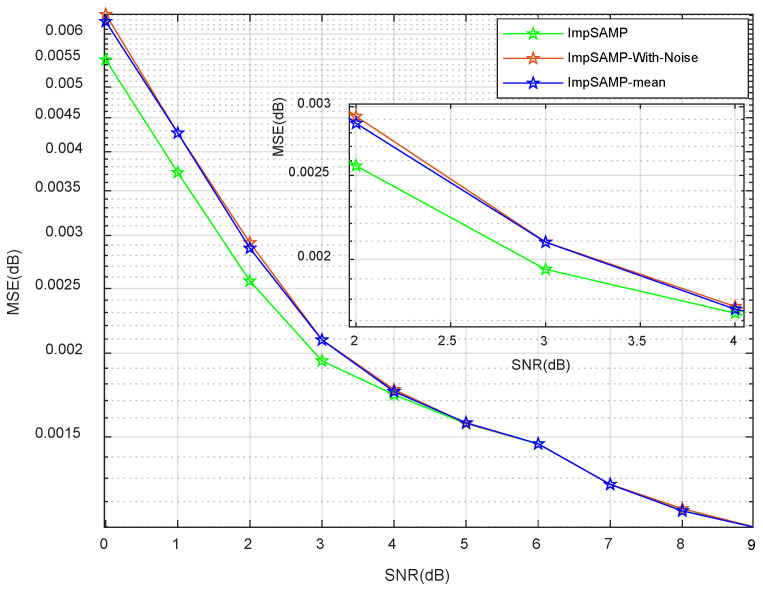
MSE vs. SNR performances of different denoising algorithms.

**Figure 4 sensors-23-06668-f004:**
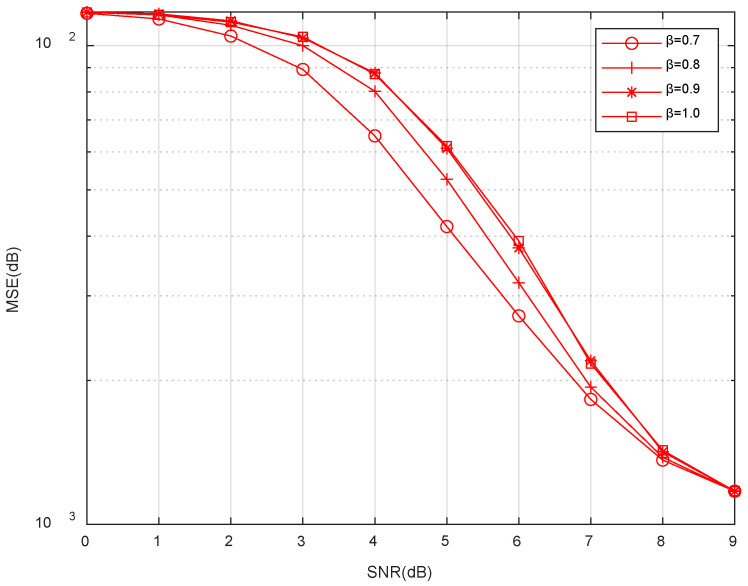
MSE vs. SNR performances of the ImpSAMP algorithm for different energy threshold coefficients, β.

**Figure 5 sensors-23-06668-f005:**
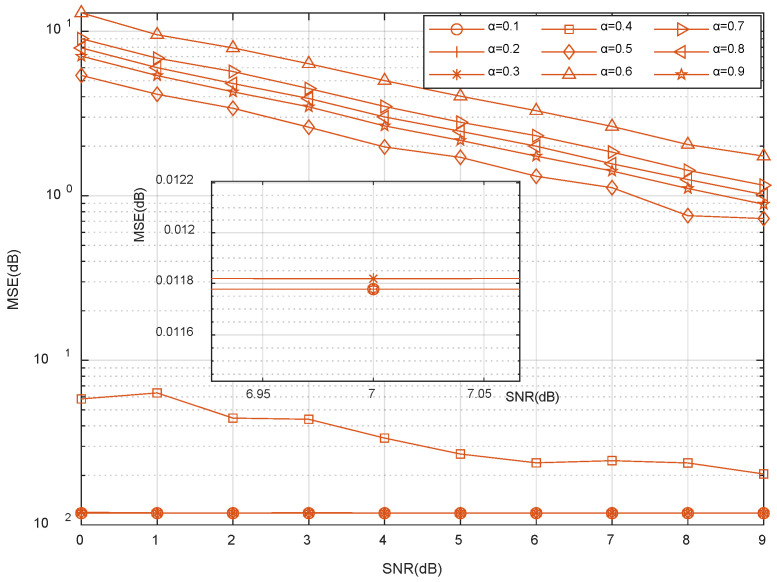
MSE vs. SNR performances of the ImpSAMP algorithm for different fuzzy threshold coefficients, α.

**Figure 6 sensors-23-06668-f006:**
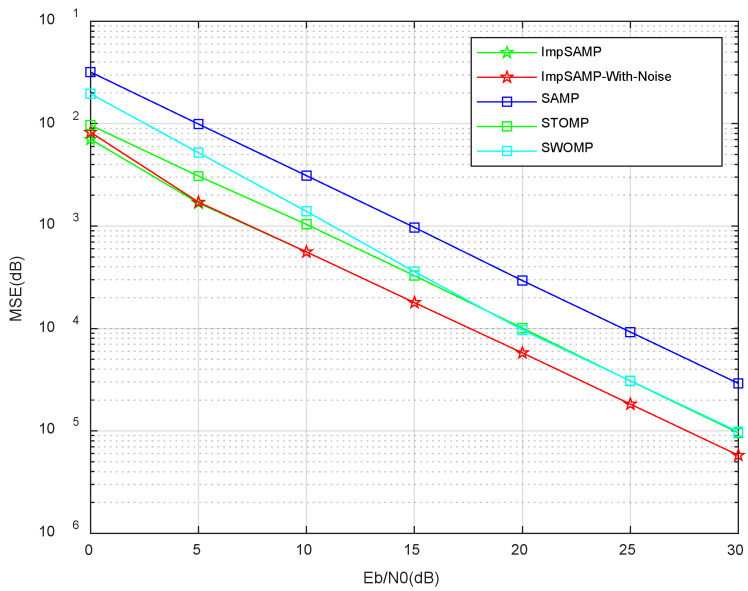
MSE vs. Eb/N0 performances of different algorithms.

**Figure 7 sensors-23-06668-f007:**
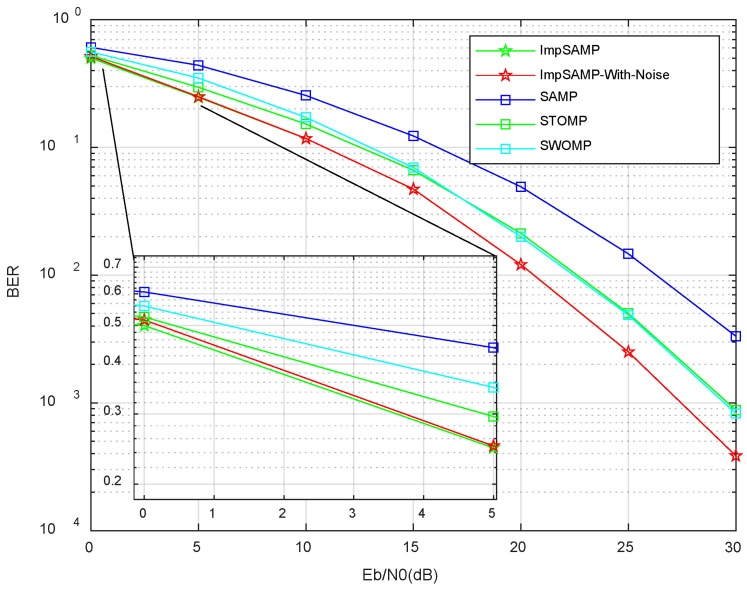
BER vs. Eb/N0 performances of different algorithms.

**Figure 8 sensors-23-06668-f008:**
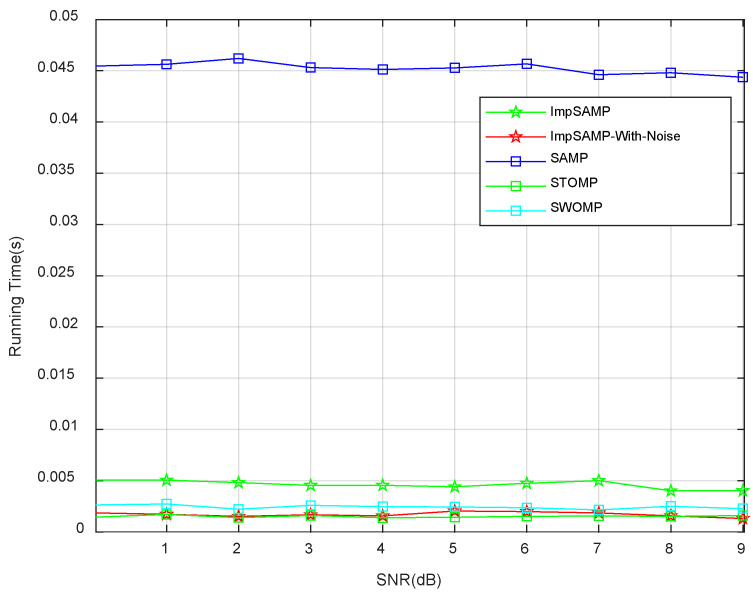
Running time vs. SNR performances of different algorithms.

**Table 1 sensors-23-06668-t001:** Simulation experiment parameters.

Parameters	Symbol	Value
Channel type	—	Rayleigh fading channel
Observation vector length	*M*	100
Sparse signal length	*N*	256
Signal sparsity	*Q*	15
Initialize the step size	$S_{0}$	4
Energy threshold coefficient	β	0.7
Fuzzy threshold coefficient	α	0.2

## Data Availability

Not applicable.

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
