# Peer review of "An Improved SAMP Algorithm for Sparse Channel Estimation in OFDM System"

_sensors, 2023, doi:10.3390/s23156668_

Round 1

Reviewer 1 Report

This paper proposes an enhancement to the so-called Sparsity Adaptive Matching Pursuit (SAMP) algorithm for channel estimation in OFDM systems, using compressed sensing. I will divide my comments into two parts: Presentation and Content.

Presentation:

The current presentation of the manuscript does not effectively retain the reader's attention due to its somewhat naïve format. For instance:

1- It lacks the professional appearance of a well-written and elegantly presented manuscript, typically produced in Latex or in recent versions of Microsoft Word with the Equation Editor utilizing Cambria Math fonts. I would recommend that the authors consider rewriting the manuscript using these tools. Overleaf, for instance, offers a free and user-friendly platform for Latex editing.

2- It is generally accepted that mathematical equations should be written as intrinsic parts of the corresponding sentence. For example:

“According to Albert Einstein's theory, energy and mass are related by

E = mc2,

where E represents the total energy in a given amount of mass m and c is the speed of light.”

The authors should consider revising numerous equations in the manuscript to adhere to this format.

Content:

1- In the Abstract, the authors state that the "Channel estimation of the Orthogonal Frequency Division Multiplexing (OFDM) system based on compressed sensing can effectively reduce the pilot overhead and enhance the utilization rate of spectrum resources." Essentially, they assert that the proposed technique can increase the spectral efficiency of the OFDM system, measured in bits/sec/Hz. Regrettably, this potentially significant result, if confirmed, was entirely overlooked in the remainder of the paper. The authors ought to present, at the very least, simulation results using channel models (like those standardized for 5G) with a Delay Spread exceeding the length of the Cyclic Prefix of the OFDM structure.

2- I recommend that the authors consider the paper by H. Ye, G. Y. Li, and B.-H. Juang, titled 'Power of Deep Learning for Channel Estimation and Signal Detection in OFDM Systems' (IEEE Wireless Communications Letters, Vol. 7, No. 1, pp. 114–117, 2018). This paper proposes a Deep Neural Network for OFDM systems with a 'zero length' Cyclic Prefix, resulting in maximum spectral efficiency.

3- How do CS techniques compare with current ML-based channel estimation and decoding?

4- Which QAM constellation was utilized in the simulations? Was it 4-PSK, 16-QAM, or 64-QAM? It's important for the authors to specify the constellation format.

5- The figure of merit used in the simulations is based on the Symbol Error Rate (SER) as a function of Signal-to-Noise Ratio (SNR), rather than the Bit Error Rate (BER) as a function of Eb/No. Using the latter would be considerably more meaningful, given that the BER threshold of 2x10-2, in current Forward Error Correction (FEC) codes, defines the region of most significant interest for performance measurement.

6- Unfortunately, the SERxSNR curves, presented in Fig. 7, appear to be well above the corresponding BER threshold of 2x10-2, as previously mentioned. It is crucial to present simulation results around this threshold, especially to investigate any potential error floor in the SERxSNR (or BERxEb/No) curves.

7- Finally, it would be intriguing to discuss potential applications in massive MIMO-OFDM, a topic that has been extensively researched for 5G and 6G systems using Machine Learning approaches.

The manuscript requires extensive editing for English language usage. For example, here are just a few typical sentences which require attention:

1- “In recent years, researchers ‘have proposed’ some signal….”

2- “The efficiency and accuracy of the GOMP algorithm ‘have been’ effectively improved.”

3- The following sentence is quite redundant and should be rewritten: “…signal 'transmitted by the transmitter' and the signal 'received by the receiver'.”

4- “In order to eliminate the influence of noise ‘on the’ received signal,…”

Reviewer 2 Report

ImpSAMP is an algorithm for channel estimation in OFDM systems. It reduces pilot overhead, improves spectrum utilization, and outperforms traditional SAMP in estimation efficiency and accuracy. I have looked at the mathematics and it looks sound. However, further elaboration on the specific enhancements, simulation results, and potential practical implications would greatly enhance its clarity and impact.

*Introduction: The introduction should include a review of the existing literature to establish the context and highlight the originality of the work.

*Motivation and Applications: Providing context regarding the motivation for reducing pilot overhead and improving spectrum resource utilization in OFDM systems is important. Highlighting specific applications or scenarios where these improvements are particularly significant would enhance the abstract's relevance and practical implications.

*Results: In the results section, it is recommended to include validation of the obtained results with the existing literature. This will help in establishing the credibility and significance of the findings.

*PAPR Consideration: Considering the Peak-to-Average Power Ratio (PAPR) issue and discussing its impact on the simulation results would be valuable. This would provide insights into the algorithm's performance in handling PAPR-related challenges.

*Drawbacks and Future Studies: Elaborating on the drawbacks of the traditional SAMP algorithm and explaining how the ImpSAMP algorithm specifically addresses them would provide a better understanding of the improvements. Additionally, suggestions for future studies to further enhance the model can be discussed.

it is fine

Reviewer 3 Report

Manuscript ID: sensors-2497457

Title:    An Improved SAMP Algorithm for Sparse Channel Estimation in OFDM System

Comments and Suggestions for Authors

The manuscript deals with a modified version of the Sparsity Adaptive Matched Pursuit Algorithm for channel estimation in a MIMO OFDM System. Authors claim that instead of fixed step size, variable step size can improve the channel estimation accuracy, in terms of MSE and BER compared to the conventional compressed sensing algorithm. Authors apply the Dice matching algorithm to improve the accuracy from fixed-step.

Following are some of the observations and comments regarding the submitted manuscript:

1.       A similar article with a similar title is recently published, however, it is not cited by the authors. Authors need to differentiate their work from the published article. Moreover, the topic is quite similar, therefore, authors have to ensure justification of their work’s novelty from previously published literature. The article is  W. Chen and M. Zhuang, "An Improved SAMP Channel Estimation Method via Compressed Sensing for MIMO-OFDM Systems," 2022 IEEE 16th International Conference on Anti-counterfeiting, Security, and Identification (ASID), Xiamen, China, 2022, pp. 59-65, doi: 10.1109/ASID56930.2022.9995751.

2.      Section II C presents the mathematical model of MIMO OFDM, which is already present in literature in numerous works, if required authors may refer to the sparse representation of the channel matrix only, instead of going into the details of the OFDM system.

3.      In Section III B the authors explain the Dice matching algorithm, however, have not given any reference that indicates the origin of the algorithm and its applications in a similar situation.

4.      Eq 25 symbol error rate can be represented in terms of baseband modulation such as QAM, QPSK etc.

5.      Table I can be reformulated in a proper manner, outlining the parameters, symbols and values in separate columns.

6.      In Table I, tap gains are seen to be fixed, do the authors assume a fixed channel condition? If varied, what type of behavior for estimation is expected?

7.      In Table II, authors have given the computational complexity of the proposed and other algorithms, however, they have not explained it in sufficient detail. Moreover, the variables defined in the table are not in equation format, therefore, it is difficult to understand these and ensure whether these are the same variables mentioned in the text before.

8.      Authors have given a summary at the end of the article, instead of a conclusion.

9.      Quality and resolution of Fig 2 & 3 is poor, and makes it difficult to read the correct values from the figures.

10.   Authors need to justify the behavior of BER in Fig 3 as it has certain peaks with SNR variation, moreover BER is quite high, can the authors explain its reason, although channel equalization is being performed? Typically it should be a waterfall curve,

Comments for Editor

The manuscript gives an improved SAMP algorithm with Dice matching algorithm for accurate channel estimation. The manuscript’s major portion is dedicated to the description of well-known concepts and equation, which can be removed to focus on the proposed technique and results. The article requires major revisions before it is accepted for publication.

Eq 25 symbol error rate can be represented in terms of baseband modulation such as QAM, QPSK etc.

The quality of English is mostly fine, a few typos are indicated in the report. 

Round 2

Reviewer 1 Report

Unfortunately, the authors seem to have misunderstood my reference to 'ML-based approaches,' which stands for machine learning-based approaches to OFDM channel estimation and detection. Additionally, the format in which they have presented the mathematical equations is somewhat unconventional.

In my opinion, further editing of the English language is still required.

Reviewer 2 Report

I am satisfied with the latest revision

readable

Author Response

Thank you for your comments and suggestions and all of your comments and suggestions are very important. They are very important to guide me in writing the manuscript, I can successfully complete the manuscript thanks to your help. Thank you very much.